# Measuring Electromagnetic Fields in Rotating Frames of Reference

**Clive C. Speake** [1,]* and **Antonello Ortolan** [2]

1  School of Physics and Astronomy, University of Birmingham, Edgbaston, Birmingham B15 2TT, UK
2  Laboratori Nazionale di Legnaro, INFN, Vialle dell'Universita 2, 3520 Legnaro, Padova, Italy; Antonello.Ortolan@lnl.infn.it
*  Correspondence: C.C.SPEAKE@bham.ac.uk

**Abstract:** We review the problem of transforming electromagnetic fields between inertial and rotating reference frames. We compare the method of straightforward tensor coordinate transformations adopted by Schiff in his well-known paper of 1939 with the method of Orthogonal Tetrads (OT) that was applied to this problem in 1964 by Irvine. Although both methods are mathematically rigorous, the transformed fields have different forms depending on the method adopted. We emphasize that the OT method is expected to predict the fields that would actually be measured by an observer in a rotating frame of reference. We briefly discuss existing experimental evidence that supports the OT approach, but point out that there appears to be little awareness in the physics community of this problem or its resolution. We use both methods to transform the electrostatic and magnetic fields generated by rotating charged spherical shells from an inertial into a co-rotating system. We also briefly describe how such an arrangement of shells could be used to measure rotation relative to the fixed stars.

**Keywords:** Special Relativity; Electromagnetism; rotating frame

## 1. Introduction

In 1939 Leonard Schiff [1] used straightforward tensor coordinate transformations to derive the form of the electrostatic and magnetic fields in rotating coordinate systems[1]. More recently Irvine and Corum [2,3] used Orthogonal Tetrads (OT's or the method of comoving frames [4,5]) to analyze this problem and obtained a different form for Maxwell's equations in rotating frames. Although relativists recognize that the OT method predicts observable physical quantities in non-inertial frames, its application to the problem of finding electromagnetic fields in rotating frames appears not to be generally acknowledged [6,7]. For example, Gopakumar et al. [8] use Schiff's results to predict electromagnetic effects in rotating frames in the context of celestial mechanics. Other papers such as [9] reference Schiff's paper but not those of Irvine or Corum. Given that Irvine's work was published more than 50 years ago and the wide use of the OT method this is surprising.

This situation is unfortunate as there are many applications where it is important to understand electromagnetism in rotating frames to be able to predict with confidence the actual fields that are measured by the non-inertial observer. The currently accepted laws of electromagnetism have been established by experiment in the rotating frame of the Earth and our current system of units is now largely based on accurate electrical measurements. Furthermore, the ubiquitous technology of electrical generators and motors, as pioneered by Michael Faraday in the 1830s, is based on devices containing

---

1  We define what we mean by 'rotating systems' in the context of each method in Section IIIC below.

rotating elements. If a theoretical framework for predicting electromagnetic physics in rotating frames were accepted (i.e., the OT method) it could allow the optimization of current, and the development of future, technologies.

The lack of clarity of the predictions of measurable electromagnetic fields in rotating frames of reference [2,3,6,7] has, arguably, led many authors of physics textbooks to avoid rigorous derivation of important physical results in favor of suggesting plausibility arguments that are misleading. An example of this is the semi-classical discussion of spin-orbit coupling that leads to the Fine Structure of atomic spectral lines which features in many undergraduate textbooks [10–12][2]. In this problem the observed splitting of atomic energy levels requires a calculation of the magnetic field observed by an electron in its orbital. This is often done using an 'instantaneous' Lorentz transformation or by simply assuming that the orbiting electron is in an inertial frame, with the nucleus producing a magnetic field due to its 'orbital' motion around it. The Thomas precession is then added to account for the fact that the electron is actually in a non-inertial frame. This analysis suggests incorrectly to the reader that the methods developed for the transformation of electromagnetic fields between inertial frames can be correctly used for cases where the frames in question are non-inertial. The authors usually acknowledge that this approach is not correct but there is never (to the authors' knowledge) mention that there exists a rigorous method for treating this problem.

Another example where the rigorous transformation of electromagnetic fields should form the basis of the discussion of a physical phenomenon is the law of Biot and Savart. This law was established experimentally in 1820 from observations of charges flowing in closed loops and, as such, the charges are in non-inertial frames. Nevertheless, it has been argued [13] that the theoretical basis for the Biot-Savart law comes again from the application of instantaneous Lorentz transformations. This reasoning leads the reader to conclude that it is the *relative* motion of the observer with respect to the rotating charges that can be used to correctly predict the magnetic field.

In this article, we review Schiff's analysis and the use of the method of Orthogonal Tetrads to transform the electrostatic and magnetic fields and find explicit expressions for the electrostatic and magnetic fields observed by inertial and co-rotating observers produced by shells carrying equal and opposite charges that are rotating about a common axis. This system was, in fact, the basis of Schiff's calculation.

We will only consider the application of Maxwell equations in vacuum (no polarizable media). The versions of Maxwell's equations that result from Schiff's ([7,14]) and Irvine's analyses, that are valid in their appropriate rotating systems, are respectively:

$$
\begin{aligned}
\nabla \cdot \vec{E} &= \frac{\rho}{\epsilon_0} + \nabla \cdot \left( (\vec{\omega} \times \vec{r}) \times \vec{B} \right) \\
\nabla \times \left( \vec{B} - \frac{1}{c^2} (\vec{\omega} \times \vec{r}) \times \vec{E} \right) - \frac{1}{c^2} \frac{\partial \vec{E}}{\partial t} &= \mu_0 \vec{j} - \frac{1}{c^2} \frac{\partial}{\partial t} \left( (\vec{\omega} \times \vec{r}) \times \vec{B} \right) \\
\nabla \times \vec{E} &= -\frac{\partial \vec{B}}{\partial t} \\
\nabla \cdot \vec{B} &= 0
\end{aligned}
\tag{1}
$$

and

$$
\begin{aligned}
\nabla \cdot \vec{E} &= \frac{\rho}{\epsilon_0} + \nabla \cdot \left( (\vec{\omega} \times \vec{r}) \times \vec{B} \right) \\
\nabla \times \vec{B} - \frac{\gamma}{c^2} \frac{\partial \vec{E}}{\partial t} &= \mu_0 \vec{j} - \frac{\gamma}{c^2} \frac{\partial}{\partial t} \left( (\vec{\omega} \times \vec{r}) \times \vec{B} \right) + \frac{\gamma^2}{c^2} \vec{B} \times (\vec{\omega} \times (\vec{\omega} \times \vec{r})) \\
\nabla \times \vec{E} + \gamma \frac{\partial \vec{B}}{\partial t} &= -\frac{\gamma}{c^2} \frac{\partial}{\partial t} \left( (\vec{\omega} \times \vec{r}) \times \vec{E} \right) + \frac{\gamma^2}{c^2} \vec{E} \times (\vec{\omega} \times (\vec{\omega} \times \vec{r})) \\
\nabla \cdot \vec{B} &= -\frac{1}{c^2} \nabla \cdot \left( (\vec{\omega} \times \vec{r}) \times \vec{E} \right)
\end{aligned}
\tag{2}
$$

---

[2] Most authors note that the full quantum mechanical treatment of this problem requires the solution of the Dirac equation.

where $\vec{\omega}$ is the rotational angular frequency and $\vec{r}$ is the radial vector distance from the rotation axis. The rotational velocity follows as $\vec{v} = \vec{\omega} \times \vec{r}$ and we define time dilation factor $\gamma$ in the usual way. We have ignored terms in Schiff's equations that are proportional to $v^2$ and higher.

These equations clearly differ from Maxwell's equations in inertial frames. They include new sources of charges and current that are proportional to the rotational speed. Most striking, perhaps, is that in the rotating system, the divergence of the magnetic field in the OT framework is non-zero. These sources are analogous to the known 'fictitious' or inertial mechanical forces also present in rotating frames of reference. However, it is important to note that the deviations from the usual form of Maxwell's equations in Equations (1) and (2) do not arise because of the inertia of any charges involved as they are independent of the inertial mass. In the following we will ignore the presence of inertial forces and this can be considered to be taking the limit where $v/c \ll 1$ or where the mass of the charged particles tends to zero.

Crater [6] considers a particle of charge, $q$, that is at rest with respect to an observer and defines the electric field, $\vec{E}$ , as the ratio of the force it experiences, $\vec{F}_1$ , due to its charge, in the limit that the charge tends to zero. We have

$$\vec{E} = \lim_{q \to 0} \frac{\vec{F}_1}{q}. \tag{3}$$

The magnetic field, $\vec{B}$, can be defined using the form of the Lorentz force in an inertial frame. Given knowledge of a particle's velocity, $\vec{v}$ , relative to the observer and the force on the moving charge, $\vec{F}_2$ , we have

$$\vec{E} + \vec{v} \times \vec{B} = \lim_{q \to 0} \frac{\vec{F}_2}{q} \tag{4}$$

It is important to note that the OT and Schiff's methods are internally consistent inasmuch as the fields predicted give Lorentz forces in the non-inertial frame that agree with those forces in the inertial frame in the limit of $v/c \ll 1$. However, the predicted fields themselves are different, so this begs the question as to what is actually *measured* by the non-inertial observer. We will calculate these electromagnetic forces in Section IV below.

We write down here relevant tensors for reference in Sections III and IV below: we will adopt the convention that the Minkowski metric tensor is diagonal with components $g_{\mu\mu} = (1, -1, -1, -1)$ and we label events in spacetime with the following coordinates $(ct, x, y, z)$. The covariant form of the Maxwell tensor can be written as a matrix in cartesian coordinates and in SI units as

$$F_{\mu\nu} = \begin{bmatrix} 0 & E_x/c & E_y/c & E_z/c \\ -E_x/c & 0 & -B_z & B_y \\ -E_y/c & B_z & 0 & -B_x \\ -E_z/c & -B_y & B_x & 0 \end{bmatrix}. \tag{5}$$

The Minkowski metric in spherical polar coordinates is

$$g_{\mu\nu} = \begin{bmatrix} 1 & 0 & 0 & 0 \\ 0 & -1 & 0 & 0 \\ 0 & 0 & -r^2 & 0 \\ 0 & 0 & 0 & -r^2 \sin^2 \theta \end{bmatrix}. \tag{6}$$

In spherical polar coordinates the covariant Maxwell tensor has the form,

$$F_{\mu\nu} = \begin{bmatrix} 0 & E_r/c & rE_\theta/c & r\sin\theta E_\phi/c \\ -E_r/c & 0 & -rB_\phi & r\sin\theta B_\theta \\ -rE_\theta/c & rB_\phi & 0 & -r^2 \sin\theta B_r \\ -r\sin\theta E_\phi/c & -r\sin\theta B_\theta & r^2 \sin\theta B_r & 0 \end{bmatrix}, \tag{7}$$

and the dual of the covariant Maxwell tensor in spherical polar coordinates is the following contravariant tensor,

$$
\mathcal{F}^{\mu\nu} = \begin{bmatrix}
0 & -B_r & -B_\theta/r & -B_\phi/r\sin\theta \\
B_r & 0 & E_\phi/cr & -E_\theta/cr\sin\theta \\
B_\theta/r & -E_\phi/cr & 0 & E_r/cr^2\sin\theta \\
B_\phi/r\sin\theta & E_\theta/cr\sin\theta & -E_r/cr^2\sin\theta & 0
\end{bmatrix}.
\tag{8}
$$

## 2. The Electrostatic and Magnetic Fields Due to a Rotating Shell of Charge Observed in an Inertial Frame

Our goal in this section is to find explicit expressions for the electrostatic and magnetic fields associated with two concentric thin, charged shells that rotate around a common axis, as measured by an inertial observer. The derivation of the fields in the interior and on the exterior of a single charged shell is given in a spherical polar coordinate system in the Appendix. Superposition of these solutions for two spherical shells of radius $R_1$ and $R_2$, with charges $Q_1$ and $Q_2$, rotating at angular rate $\Omega$ gives the fields within three regions:

For region *a* where $r < R_1$:

$$
\vec{E}_a = 0,
\tag{9}
$$

and

$$
\vec{B}_a = \frac{\mu_0}{2\pi}\left(\frac{m_1}{R_1^3} + \frac{m_2}{R_2^3}\right)\left(\cos\theta\hat{r} - \sin\theta\hat{\theta}\right).
\tag{10}
$$

For region *b* where $R_1 < r < R_2$

$$
\vec{E}_b = \frac{Q_1}{4\pi\epsilon_0 r^2}\hat{r},
\tag{11}
$$

$$
\vec{B}_b = \frac{\mu_0}{2\pi}\left(\left(\frac{m_1}{r^3} + \frac{m_2}{R_2^3}\right)\cos\theta\hat{r} + \left(\frac{m_1}{2r^3} - \frac{m_2}{R_2^3}\right)\sin\theta\hat{\theta}\right).
\tag{12}
$$

For region *c* where $r > R_2$

$$
\vec{E}_c = \frac{(Q_1 + Q_2)}{4\pi\epsilon_0 r^2}\hat{r},
\tag{13}
$$

and

$$
\vec{B}_c = \frac{\mu_0\,(m_1 + m_2)}{2\pi r^3}\left(\cos\theta\hat{r} + \frac{\sin\theta}{2}\hat{\theta}\right),
\tag{14}
$$

where the magnetic moment of the *i*th shell is given as $m_i = \frac{\Omega Q R_i^2}{3}$. As expected, at the surfaces of the shells the radial components of the magnetic fields are continuous and the difference between the inner and outer polar fields is equal to the product of the total charge at that latitude and the frequency of rotation of the shell. In what follows we will consider the case where the shells have equal and opposite charges but the above equations give the more general solution.

## 3. Transformation of the Fields

*3.1. Special Relativity*

It will be useful for what follows to compare the transformations of electromagnetic fields between inertial frames (using Lorentz transformations) with the analysis of Schiff and the Orthogonal Tetrad method. We can imagine that we have a reference inertial frame in which the covariant Maxwell tensor given in Equation (5) is valid and that, working in a cartesian coordinate system, we have another frame moving at speed $v$ in the $\hat{x}$ direction. The covariant Lorentz transformation tensor can be written in matrix form as

$$S_{\mu\nu} = \begin{bmatrix} \gamma & \gamma\beta & 0 & 0 \\ \gamma\beta & \gamma & 0 & 0 \\ 0 & 0 & 1 & 0 \\ 0 & 0 & 0 & 1 \end{bmatrix}, \tag{15}$$

where we define $\beta$ and $\gamma$ in the usual way in terms of the linear speed. We can transform the Maxwell tensor into the moving frame using the standard techniques

$$F'_{\mu\nu} = S_{\mu\alpha} S_{\nu\beta} F_{\alpha\beta}, \tag{16}$$

which gives the transformed version of the Maxwell tensor,

$$F'_{\mu\nu} = \begin{bmatrix} 0 & E_x/c & \gamma/c\left(E_y - vB_x\right) & \gamma/c\left(E_z + vB_y\right) \\ -E_x/c & 0 & -\gamma\left(B_z - vE_y/c^2\right) & \gamma\left(B_y + vE_z/c^2\right) \\ -\gamma/c\left(E_y - vB_x\right) & \gamma\left(B_z - vE_y/c^2\right) & 0 & -B_x \\ -\gamma/c\left(E_z + vB_y\right) & -\gamma\left(B_y + vE_z/c^2\right) & B_x & 0 \end{bmatrix}. \tag{17}$$

Comparison of Equation (17) with Equation (5) gives the expected transformations [15] between the fields:

$$E'_x = E_x, \quad E'_y = \gamma\left(E_y - vB_z\right), \quad E'_z = \gamma\left(E_z + vB_y\right), \tag{18}$$

and

$$B'_x = B_x, \quad B'_y = \gamma\left(B_y + vE_z/c^2\right), \quad B'_z = \gamma\left(B_z - vE_y/c^2\right). \tag{19}$$

It is clear that for the case of two reference frames in rectilinear motion, Lorentz transformations can be used to predict the measured values of the electrostatic and magnetic fields observed in one frame in terms of the fields measured in the other. Each observer can predict the results of measurements made by an observer in another reference frame given the measurement she or he makes in her or his own frame and their relative velocity. The idea that Special Relativity is based on the measurements of electromagnetic fields made by observers in relative states of motion was central to Einstein's original paper in 1905 [16].

### 3.2. Schiff's Method

Schiff motivated his paper with a question that was asked by Oppenheimer: suppose that we have two thin shells, as described in Section II, with equal and opposite charges. If these shells are not rotating, we would expect that a non-rotating observer situated outside the outer shell would measure no electrostatic or magnetic field. However if the shells rotate the observer would expect to measure a magnetic field but no electrostatic field. These fields are given by Equations (13) and (14) with $Q_1 = -Q_2$. We now imagine what would be measured if the observer is rotating and the shells are not. A naive view would be that the shells appear to be undergoing rotation relative to the observer and therefore a magnetic field would be measured. This response would be expected following the logic applied to solve some well-known physics problems in textbooks, as was described in the introduction. The apparent paradox is that we expect that this field, observed from a rotating coordinate system, would naturally be found by a transformation from the fields measured by a non-rotating observer. However, we know that this inertial observer measures no electrostatic or magnetic fields. This problem is referred to as Schiff's paradox. Schiff then proceeded to derive the electromagnetic fields in the rotating coordinate system using a coordinate tensor transformation. There has been much discussion in the literature, [2,3,6,7], about the choice of the 'correct' form of the Maxwell tensor for the transformation into the rotating coordinate system. It could be either covariant or contravariant with different results being obtained for each case. However, there is no such ambiguity for transformations between inertial frames as demonstrated by the well-known results of Section III A. However, it is argued by Crater [6] that the covariant 4-velocity is not a valid quantity and this leads to the choice of

the covariant form of the Maxwell tensor as is shown when we calculate the forces predicted in the rotating frame in Section IV below. This was the option taken by Schiff.

The covariant and contravariant Galilean transformations into a coordinate system that is rotating at an angular velocity $\omega$ are, respectively,

$$\Lambda_{\mu\nu} = \begin{bmatrix} 1 & 0 & 0 & 0 \\ 0 & 1 & 0 & 0 \\ 0 & 0 & 1 & 0 \\ \omega/c & 0 & 0 & 1 \end{bmatrix}, \tag{20}$$

and

$$\Lambda^{\mu\nu} = \begin{bmatrix} 1 & 0 & 0 & 0 \\ 0 & 1 & 0 & 0 \\ 0 & 0 & 1 & 0 \\ -\omega/c & 0 & 0 & 1 \end{bmatrix}. \tag{21}$$

These follow from the transformation to the azimuthal angle $\phi'$ in the rotating coordinate system from the angle $\phi$ in the inertial frame. The contravariant metric, for example, for the rotating coordinate system can then be calculated using

$$g_{Gal}^{\mu\nu} = \Lambda^{\mu\alpha} \Lambda^{\nu\beta} g_{\alpha\beta} \tag{22}$$

where $g_{\alpha\beta}$ is given in Equation (6). We find

$$g_{Gal}^{\mu\nu} = \begin{bmatrix} 1 & 0 & 0 & -\omega/c \\ 0 & -1 & 0 & 0 \\ 0 & 0 & -1/r^2 & 0 \\ -\omega/c & 0 & 0 & -1/\gamma^2 r^2 \sin^2\theta \end{bmatrix}. \tag{23}$$

For the rotating frame we now define, in spherical polar coordinates,

$$\beta = \frac{v}{c} = \frac{r\omega\sin\theta}{c}, \tag{24}$$

and

$$\gamma = \left(1 - \beta^2\right)^{-1/2}, \tag{25}$$

with $\vec{v} = r\omega\sin\theta\,\hat{\phi}$. We find that the Maxwell tensors are then transformed into the following form in the rotating coordinate system,

$$F'_{\mu\nu} = \begin{bmatrix} 0 & (E_r - r\sin\theta B_\theta\omega)/c & r(E_\theta + r\sin\theta B_r\omega)/c & r\sin\theta E_\phi/c \\ -(E_r - r\sin\theta B_\theta\omega)/c & 0 & -rB_\phi & r\sin\theta B_\theta \\ -r(E_\theta + r\sin\theta B_r\omega)/c & rB_\phi & 0 & -r^2\sin\theta B_r \\ -r\sin\theta E_\phi/c & -r\sin\theta B_\theta & r^2\sin\theta B_r & 0 \end{bmatrix}, \tag{26}$$

and

$$\mathcal{F}'^{\mu\nu} = \begin{bmatrix} 0 & -B_r & -B_\theta/r & -B_\phi/r\sin\theta \\ B_r & 0 & E_\phi/cr & -E_\theta/cr\sin\theta - B_r\omega/c \\ B_\theta/r & -E_\phi/cr & 0 & E_r/cr^2\sin\theta - B_\theta\omega/rc \\ B_\phi/r\sin\theta & E_\theta/cr\sin\theta + B_r\omega/c & -E_r/cr^2\sin\theta + B_\theta\omega/rc & 0 \end{bmatrix}. \tag{27}$$

Comparison of Equations (7) with (26) or (8) and (27) gives the transformations

$$E'_r = E_r - vB_\theta, \quad E'_\theta = E_\theta + vB_r, \quad E'_\phi = E_\phi, \tag{28}$$

which can be written

$$\vec{E}' = \vec{E} + \vec{v} \times \vec{B}. \tag{29}$$

We also have

$$\vec{B}' = \vec{B}. \tag{30}$$

These transformations agree with the results calculated by Schiff, although they are not explicitly stated in Schiff's paper and Schiff works in cartesian coordinates. The full set of Maxwell's equations was given in the introduction in the rotating coordinate system and has been derived by many authors [7,14], and contain fictitious sources of current and charge that appear due to the rotation of the observer's coordinate system. The resolution to Schiff"s 'paradox', as far as Schiff was concerned (see Section IIIC below), was that the current flow that would be expected to be observed in the rotating system according to Special Relativistic instincts (the 'instantaneous application of Lorentz transformations') is cancelled by a fictitious current density that exists in the rotating coordinate system. Alternatively, all observers can agree that the magnetic field is invariant under the transformation into the rotating coordinate system. We must be, again, careful to state that there is no physical observer in the rotating coordinate system.

Equations derived in Section II can be used to calculate explicit expressions for the fields due to the charged shells in the rotating coordinate system. It is interesting to note that Schiff used a perturbation technique to solve this problem; however, Corum [3,17], found the solution using the transformations between the fields in the two coordinate systems given in Equations (28)–(30). Considering the simple case where the total charge on each shell is equal but of opposite polarity, we obtain the following fields in the three regions:

For $r < R_1$,

$$\vec{E}'_a = \frac{\mu_0 v}{6\pi} Q\Omega \left( \frac{1}{R_1} - \frac{1}{R_2} \right) \hat{\varrho}, \tag{31}$$

$$\vec{B}'_a = \vec{B}_a; \tag{32}$$

For $R_1 < r < R_2$,

$$\vec{E}'_b = \frac{Q}{4\pi\epsilon_0 r^2} \hat{r} - \frac{\mu_0 v}{6\pi} Q\Omega \left( \left( \frac{R_1^2}{2r^3} + \frac{1}{R_2} \right) \sin\theta \hat{r} - \left( \frac{R_1^2}{r^3} - \frac{1}{R_2} \right) \cos\theta \hat{\theta} \right), \tag{33}$$

$$\vec{B}'_b = \vec{B}_b \tag{34}$$

and for $r > R_2$,

$$\vec{E}'_c = \frac{\mu_0 v}{6\pi r^3} \left( R_1^2 - R_2^2 \right) Q\Omega \left( -\frac{\sin\theta}{2} \hat{r} + \cos\theta \hat{\theta} \right), \tag{35}$$

$$\vec{B}'_c = \vec{B}_c, \tag{36}$$

where we denote a unit vector in the cylindrical polar direction as $\hat{\varrho}$. It is interesting to note that in this analysis, there *are* electric fields within the inner shell and outside the external shell, both of which are not expected according to the laws of electromagnetism in inertial frames.

### 3.3. The Method of Orthogonal Tetrads

The fundamental problem with dealing with physics in rotating coordinate systems is that the Special Relativistic time dilation factor is not uniform across the rotating system due to the rotational speed being proportional to the distance from the axis of rotation. This problem is referred to anholonomicity and it is this property of the coordinate system that gives rise to the Thomas Precession and the Sagnac effect [2,3]. The OT method is often used in General Relativity to calculate quantities that are measured by physical observers in non-inertial frames.

Central to Irvine's application of the OT method was the concept of the observer's reference frame which, like the Lorentz frame of Special Relativity, has spatial axes that are mutually orthogonal and orthogonal to the tangent of the world line of the observer with a diagonal metric as in Equation (6). An orthonormal set of basis vectors are thus defined. Clearly the metric of the rotating coordinate system given in Equation (23) is not diagonal. Schiff's analysis therefore derives a form of the electromagnetic fields in the *rotating coordinate system*, but this is not the frame of a physical observer and therefore the fields cannot therefore be considered to be observable. As with Special Relativity the problem of the selection of the covariant or contravariant form of the Maxwell tensor does not arise in the OT method as there is no real distinction between these in a frame described by Orthogonal Tetrads. A system of reference is selected in an inertial frame and scale factors defined by a matrix $\underline{\lambda}^i_{(a)}$ are chosen that can be used to convert local tensor quantities into their tetrad scalar components. The index in brackets refers to the component in the OT frame, known as the Lorentz index, whereas the other is the tensor index that has been used elsewhere in this paper. For example, if we work in a specific spherical polar coordinate system in an inertial frame, we would transform the Maxwell tensor given in Equation (A8) using the following matrix,

$$\underline{\lambda}^i_{(a)} = \begin{bmatrix} 1 & 0 & 0 & 0 \\ 0 & 1 & 0 & 0 \\ 0 & 0 & 1/r & 0 \\ 0 & 0 & 0 & 1/r\sin\theta \end{bmatrix}. \tag{37}$$

We can then determine the form of the Maxwell tensor in the OT inertial frame as

$$F_{(ab)} = \underline{\lambda}^i_{(a)}\underline{\lambda}^j_{(b)}F_{ij}, \tag{38}$$

or, writing the components as a matrix, we find,

$$F_{(ab)} = \begin{bmatrix} 0 & E_r/c & E_\theta/c & E_\phi/c \\ -E_r/c & 0 & -B_\phi & B_\theta \\ -E_\theta/c & B_\phi & 0 & -B_r \\ -E_\phi/c & -B_\theta & B_r & 0 \end{bmatrix}. \tag{39}$$

which recalls Equation (5). To define a reference frame of an observer moving with respect to this inertial system we need to associate with that observer an orthonormal set of spacetime coordinate axes. This is done by solving a set of differential equations, (the Frenet-Serret equations), that match the time axis of the OT frame with the 4-velocity of the observer. This process involves finding the derivatives of the Christoffel symbols in the coordinate system used. This results in a transformation that is essentially an instantaneous Lorentz transformation. We can transform a covector, for example, in the $\underline{\lambda}^i_{(a)}$ OT system using the matrix

$$L^{(a)}_{\ (b)} = \begin{bmatrix} \gamma & 0 & 0 & \gamma\beta \\ 0 & 1 & 0 & 0 \\ 0 & 0 & 1 & 0 \\ \gamma\beta & 0 & 0 & \gamma \end{bmatrix}. \tag{40}$$

Notice the similarity of this to Equation (15) but with $\gamma$ and $\beta$ now defined in Equations (24) and (25). We can find the OT form of the Maxwell tensor in the reference frame that accompanies the observer as

$$F'_{(cd)} = L^{(a)}_{\ (c)}L^{(b)}_{\ (d)}F_{(ab)}, \tag{41}$$

and this is analogous to the doubly covariant tensor transformation leading to Equation (16) and it is not, therefore, surprising that the result is analogous to Equation (17). We can then find the

following relationships between the electric field in the inertial frame and the rotating reference frame of an observer,

$$E'_r = \gamma \left( E_r - vB_\theta \right), \quad E'_\theta = \gamma \left( E_\theta + vB_r \right), \quad E'_\phi = E_\phi. \tag{42}$$

Comparison with Equation (28) shows that the $\hat{r}$ and $\hat{\theta}$ components of the electric fields predicted by the OT method differ from those predicted by Schiff's method only by a factor of $\gamma$. The explicit expressions for the electric fields due to the rotating shells can therefore be easily found from Equations (9)–(14). In contrast to Schiff's method, the OT method predicts that the magnetic field no longer remains unchanged in the rotating system. We can write,

$$B'_r = \gamma \left( B_r + vE_\theta / c^2 \right), \quad B'_\theta = \gamma \left( B_\theta - vE_r / c^2 \right), \quad B'_\phi = B_\phi. \tag{43}$$

The relationships given in Equations (42) and (43), are formally identical with the standard results of Special Relativity as derived in (17) but the components of the electromagnetic fields are now given in terms of the spherical polar rather than cartesian coordinates and the parameter $\gamma$ is now in terms of the rotational speed rather than a linear one. The spherical polar coordinates are defined in the reference system in the inertial frame. To derive the form of Maxwell's equations given in (2) we need to express the fields and sources in terms of a coordinate system that co-rotates with the non-inertial observer (local coordinates). This is accomplished by first transforming $F_{\mu\nu}$ in Equation (5) into the rotating coordinate system to find Equation (26), in the way employed by Schiff. Now we use another transformation, which is again based on the Frenet-Serret equations, to obtain the OT components of the fields. This gives the following transformations,

$$E'_{or} = E'_r, \quad E'_{o\theta} = E'_\theta, \quad E'_{o\phi} = \gamma^2 E'_\phi, \tag{44}$$

and

$$B'_{or} = B'_r + \gamma \frac{v^2}{c^2} B_r, \quad B'_{o\theta} = B'_\theta + \gamma \frac{v^2}{c^2} B_\theta, \quad B'_{o\phi} = B'_\phi, \tag{45}$$

where we denote the fields in the observer's frame measured with local coordinates with the subscript $o$ and have defined them in terms of the fields defined in Equations (42) and (43) which are the field measured with the inertial coordinates. We see that to first order in $v/c$ the fields in each of these frames are equal. We refer the reader to the original papers [2,17] for more details of the OT method and we note also that Corum used the OT method to derive equations for the fields due to a charged shell observed by a co-rotating observer using the coordinates defined in the inertial reference frame [17].

In the case of the rotating shells with equal and opposite charges it is only in the region between the shells where there exists a finite electric field, so only the magnetic field in this region is modified. To lowest order in $v/c$ we find

$$\vec{B}'_b = \vec{B}_b - \frac{\mu_0 Q \omega \sin\theta}{4\pi r} \hat{\theta}, \tag{46}$$

where $\vec{B}_b$ is defined in Equation (12).

## 4. Force on a Charge

### 4.1. Special Relativity

If we have a point charge of value $q$ moving in the reference frame described in Section IIIA with a 4-velocity

$$u^\mu = \left( \begin{array}{cccc} c, & u_x, & u_y, & u_z \end{array} \right), \tag{47}$$

the 4-force, $f^\mu$, can be calculated from the relation

$$f^\mu = q\, g^{\mu\nu} F_{\nu\alpha} u^\alpha, \tag{48}$$

giving

$$f^\mu = \left( \vec{f} \cdot \vec{u}/c, \vec{f} \right), \tag{49}$$

where the 3-vector force experienced by the charge is, of course, given by the Lorentz force,

$$\vec{f} = q \left( \vec{E} + \vec{u} \times \vec{B} \right), \tag{50}$$

and therefore,

$$\vec{f} \cdot \vec{u} = q \left( E_x u_x + E_y u_y + E_z u_z \right). \tag{51}$$

Using the contravariant form of the Lorentz transformation of Equation (15) and the 4-velocity in the inertial frame, Equation (47), we can find the 4-velocity in the moving reference frame,

$$u'^\mu = \gamma' \left( c, \quad \vec{u}' \right), \tag{52}$$

where

$$\gamma' = \gamma \left( 1 - \frac{\beta u_x}{c} \right). \tag{53}$$

The 3-velocity then becomes

$$\vec{u}' = \left( \frac{u_x - v}{1 - \beta\beta_x}, \quad \frac{u_y}{\gamma'}, \quad \frac{u_z}{\gamma'} \right). \tag{54}$$

where $\beta_x = u_x/c$. The relationship between the 3-force, $\vec{f}'$, and the 4-force, $f'^\mu$ in the moving reference frame is,

$$f'^\mu = \left( \gamma' \vec{f}' \cdot \vec{u}/c, \quad \gamma' \vec{f}' \right), \tag{55}$$

and we can use Equation (48), expressed in terms of the Maxwell tensor and the 4-velocity in the moving frame, to calculate the 4-force in the moving frame. We find the usual transformations for the 3-force as given, for example, by French [18]. In the limit $v \ll c$, $\vec{f}' = \vec{f}$, we can conclude that the forces measured in both frames are equal. This illustrates a key result of Special Relativity, that the two observers measure different magnetic and electrostatic fields in their respective frames but when they use the universal physics of electromagnetism, they agree on the force.

### 4.2. Schiff's Method

Again, adopting Schiff's definition of the physically meaningful Maxwell tensor as being the covariant one, we can calculate the covariant 4-force on a charge, $q$, in the inertial frame by again using Equation (48). The 4-velocity of a particle in the inertial frame expressed in spherical polar coordinates is the contravariant vector,

$$u^\mu = \left( c, u^r, u^\theta, u^\phi \right). \tag{56}$$

with $u^r = \dot{r}$, $u^\theta = \dot{\theta}$ and $u^\phi = \dot{\phi}$. We find the 4-velocity in the rotating coordinate system as

$$u'^\mu = \left( c, u^r, \dot{\theta}, \dot{\phi} - \omega \right). \tag{57}$$

Using Equation (7) and, if we restrict ourselves in the interests of simplicity to the $\phi$ component of the velocity, we find that the contravariant 4-force in the inertial frame is

$$f^\mu = q \left( r \sin\theta E_\phi \dot{\phi}/c, (E_r - r\sin\theta\dot{\phi}B_\theta), (E_\theta + r\sin\theta\dot{\phi}B_r) / r, E_\phi/r\sin\theta \right) \tag{58}$$

The covariant 4-force on a charge as calculated in the rotating coordinate system is given by

$$f'^\mu = qg' F'_{\alpha\beta} u'^\beta, \tag{59}$$

with the covariant Maxwell tensor being given by Equation (26) and the transformed contravariant metric tensor being given by Equation (23). We find

$$f'^{\mu} = q \left( r \sin \theta E_{\phi} \dot{\phi}/c, (E_r - r \sin \theta \dot{\phi} B_{\theta}), (E_{\theta} + r \sin \theta \dot{\phi} B_r,)/r, E_{\phi}/r \sin \theta \left(1 - \frac{r^2 \sin \theta^2 \omega \phi}{c^2}\right) \right). \quad (60)$$

Thus, in the limit of $v \ll c$ the 4-force, and therefore the 3-force, in the inertial and the rotating coordinate system are equal. Notice that when dealing with the forces in the spherical polar coordinate system that the dimensions of the components are not the same. We conclude that, as was the case in Special Relativity, despite the different form taken by the electrostatic field in the rotating frame (see Equations (29) and (30)), the forces are equal.

*4.3. The Method of Orthogonal Tetrads*

The force on a moving charge can be calculated using Equation (48), which can be written in the OT formalism as [2],

$$f_{(a)} = q F_{(ab)} u^{(b)}. \quad (61)$$

In the inertial frame the 4-velocity is

$$u^{(a)} = \lambda_{\mu}^{(a)} u^{\mu} = \left[c, \dot{r}, r\dot{\theta}, r \sin \theta \dot{\phi}\right], \quad (62)$$

which we can transform into the non-inertial frame using the matrix representing the Lorentz transformation,

$$u'^{(a)} = L^{(a)}{}_{(b)} u^{(b)} = \gamma' \left[c, u'^j\right]. \quad (63)$$

We find that $u'^{(a)}$ has the same form as Equation (52) but now we can define

$$\gamma' = \gamma \left(1 - \beta \frac{r \sin \theta \dot{\phi}}{c}\right), \quad (64)$$

and the components of the 3-velocity can be written

$$\vec{u}' = \left[\frac{\dot{r}}{\gamma'}, \frac{r\dot{\theta}}{\gamma'}, \frac{r(\dot{\phi} - \omega)}{(1 - \beta\beta_{\phi})}\right], \quad (65)$$

with $\beta_{\phi} = r \sin \theta \dot{\phi}/c$. These equations are analogous to those in Section IV A. The 4-force follows in an obvious way when we use the transformed version of the Maxwell tensor written in component form in the non-inertial frame given in Equation (41). Therefore, the OT method predicts that in the limit of $\beta \ll c$, the 4-force on the moving charge in the non-inertial frame is in agreement with that in the inertial frame.

## 5. Discussion

We have seen that in Special Relativity, Schiff's analysis and the OT method, in the limit of $v/c \ll 1$, all observers agree on the electromagnetic force experienced by charges given the respective transformation laws. Furthermore, the magnetic and electrostatic fields in the moving frame (moving either with linear relative velocity or angular velocity relative to the fixed stars) differ from the fields observed in the inertial reference frames. It could be claimed that the story ends there and that there is no point in finding Maxwell's equations in the rotating system as we can use the simpler physics in the inertial frame to get the same answer! However, we should expect to be able to predict not only agreement between the forces measured but also the correct fields *measured* by the observers. Using this criterion, we would choose the OT method as being satisfactory.

We have noted that according to the OT method, the rotating observer who is external to *both* shells will only measure the magnetic field generated by their rotation relative to the fixed stars. This

has been pointed out by Corum [3]. Schiff showed, following his approach, that the magnetic field of a *single* charged shell would depend only on the rotational motion of that shell and not the rotational motion of the observer. However, unlike Schiff's calculation, the OT method introduces new magnetic fields which, in the limit of low speeds, are given by Equation (43) and can be written as

$$\vec{B}' = \vec{B} - \frac{1}{c^2} \left( \vec{v} \times \vec{E} \right), \tag{66}$$

which resembles the usual Lorentz transformation. It is easily shown that the divergence of Equation (66) is zero for the case of linear motion but in the case of rotation we have

$$\nabla \cdot \vec{B}' = -\frac{2}{c^2} \vec{E} \cdot \vec{\omega}. \tag{67}$$

We must be careful to note that this divergence is in terms of the inertial coordinate system and not the local coordinates of the observer in the rotating frame.

Marsh [19] showed that the magnetic field, $\vec{B}_s$, measured by an inertial observer, due to a simple circular loop of rotating charge, expressed in terms of a cylindrical polar coordinate system, is

$$\vec{B}_s = +\frac{1}{c^2} \left( (\vec{\omega} \times \vec{r}) \times \vec{E} + \vec{\omega}\Phi - \omega(z - z')\vec{E} \right), \tag{68}$$

where $\Phi$ is the electrostatic potential, $z - z'$ is the axial distance between the ring and the position of the measurement and $\vec{\omega}$ is the angular rotation rate of the ring. The first term is in the form of the inverse Lorentz transformation at low speed of the electrostatic field but with opposite sign, as given in Equation (66), and the other terms arise due to the integration of this field around a closed loop. The divergence of Equation (68) is zero as is to be expected. The magnetic field that would be predicted by the OT method in the rest frame of a co-rotating observer due to such a ring would be the Lorentz transformed version of the electrostatic field at that location and would not be given by Equation (68). It is not surprising therefore that there is not a complete cancellation of the magnetic fields in between the shells in Equation (46). It would therefore be possible for an observer located there to measure a field that would depend on the rotation rate of the shells and that of the observer. The magnetic field measured by an observer that co-rotates with a rotating source of charge with axial symmetry is therefore not zero as would be expected if the law of Biot and Savart depended only on the relative motion of the observer as suggested by Jackson [13] and mentioned in Section I. If the magnetic field is produced by a current flow, with no net electric field in the rest frame of the current, clearly the field will not depend on the motion of the observer.

In the case of Schiff's calculation, it was only the electric field that transformed between the inertial and rotating systems. It is well known [20] that the Lorentz force acting on a charge that is stationary in a rotating frame gives rise to a 'fictitious' electrostatic field that manifests itself within metallic objects that rotate within a magnetic field. Thus, the field in the rotating system can indeed be considered to have non-zero divergence even in a vacuum. This is indeed the principle behind Faradays unipolar generators. Equation (31) shows that there is an electrostatic field within an inertially rotating charged sphere when measured in a rotating frame. This is also a surprising result which is also predicted by the OT method.

Given the experimental evidence, it is unlikely that there could be an error in the prediction of the Fine Structure of atomic spectra through the application of an 'instantaneous' Lorentz transformations, as was mentioned in our Introduction. Schiff's method contradicts this prediction but the OT method supports it as the magnetic field seen by the electron is indeed due the Lorentz transformation of the electric field due to the nucleus given by Equation (66). In this case, the nucleus must be considered to be a point charge and not a charge loop and again the divergence of this magnetic field is not zero. The electron also carries a coordinate system around with it that does not coincide with the inertial system

of reference that was used in the calculations in this paper and, when this is included, the Thomas precession is predicted by the OT method.

## 6. Conceptual Design of Experimental Tests

There have been several authors who have described designs [21,22] and the performance of electromagnetic devices [23,24] that can measure rotation with respect to the fixed stars. These devices can be referred to as electromagnetic gyroscopes and rely on the physical principles outlined in this paper. However, none of these ideas, to the author's knowledge, have been discussed in terms of the theoretical predictions described here. Suppose that an experimenter wishes to exploit the charged shells, described above, to measure the rotation of his laboratory with respect to the fixed stars. The first thing to note is that the fields and charges measured by the inertial observer are not directly available to this experimenter. The charges could be measured in the rotating laboratory frame using devices that would measure the electrostatic fields, $\vec{E}'$, that they produce, for example. However, the results of such measurements would be transformed versions of the electrostatic fields observed in the inertial frame, $\vec{E}$, as given in Equation (42), for example. To relate the charge to the measured field, $\vec{E}'$, using the form of Maxwell's equations in the inertial frame, a reverse transformation would need to be applied or Maxwell's equations in the rotating frame would need to be solved. This process would itself require a knowledge of the rotation rate of the observer with respect to the fixed stars. It appears that there is a calibration issue without a simple solution but the seriousness of this problem would depend on the target accuracy.

We see that in the inertial frame there appear magnetic fields, due to the rotation of the charged spherical shells, within the inner shell (see Equations (9) and (10)) and that these fields will be directly measurable in the experimenter's frame only in the case where the shells are rotating with respect to the fixed stars, irrespective of the rotational motion of the observer and the exact magnitude of the charges. So, such a device could be the basis of an electromagnetic gyroscope.

## 7. Conclusions

We have compared Schiff's results for the electromagnetic fields in a rotating frame with those of the method of Orthogonal Tetrads (OT). We give specific results for the case of two rotating shells carrying charges of equal magnitude but opposite polarity. The two theories appear to provide conflicting results but this discrepancy can be understood when we realize that fields predicted by Schiff's calculation are not fields that can be measured by a physical observer. We note that there is essentially no acknowledgement within the physics community of the results of the OT method for handling this important physical problem. This is evident in the non-rigorous way well-established textbooks treat problems such as the energy shifts introduced due to the coupling of spin and angular momenta in elementary atomic physics. We note that Schiff's method apparently contradicts the well-accepted result derived for this problem but note also that Irvine's OT method is consistent with it. However, the relativity of the law of Biot and Savart *is* called into question by the OT method.

We suggest that experimental effort should be made to examine the predictions of the form of the magnetic and electrostatic fields in a rotating frame of reference and have briefly considered how this can be done. We have argued that the results of this work would be useful in establishing the relativity of electromagnetism and this would have technological and academic/pedagogical implications. Of particular interest is that within the non-inertial frame using local coordinates, the magnetic field has a finite divergence.

**Author Contributions:** C.C.S. devised the project and performed the analytic calculations. A.O. was involved in the development of the theoretical framework. Both the authors discussed the results and contributed to the final version of the manuscript. All authors have read and agreed to the published version of the manuscript.

**Funding:** C.C.S. would like to thank STFC, UK grant number ST/S000305/1 and University of Birmingham for support.

**Acknowledgments:** We would like to thank Robin Tucker of the University of Lancaster, Chris Moore, Will Farr of the University of Birmingham and Giovanni Carugno and Giuseppe Ruoso of the University of Padova for useful discussions. We also thank Ted Forgan, Chris Muirhead, Alastaire Rae and Joe Vinen of University of Birmingham for encouragement and stimulating ideas.

**Conflicts of Interest:** The authors declare no conflict of interest.

## Appendix A

We suppose that we have one thin shell of radius $R$ that has a total charge $Q$ uniformly distributed across its surface. The electrostatic fields interior and exterior to the shell can be found using Gauss's theorem,

$$\vec{E}_{in} = 0, \tag{A1}$$

and

$$\vec{E}_{out} = \frac{Q}{4\pi\varepsilon_0 r^2}\hat{r}. \tag{A2}$$

The general solutions to Laplace's equation in spherical polar coordinates in terms of the magnetic vector potential, $\vec{A}$, and constants $a$ and $b$ are

$$\vec{A}_{in} = ar\,\sin\theta\hat{\phi}, \tag{A3}$$

$$\vec{A}_{out} = \frac{b}{r^2}\sin\theta\hat{\phi}. \tag{A4}$$

Using the continuity of the radial component of the magnetic field at the shell, we find $b = aR^3$. Using Ampere's law for the polar component of the field, we find

$$\frac{b}{R^3} + 2a = \sigma\mu_0 R\Omega, \tag{A5}$$

where $\sigma$ is the surface charge density of the shell. We can therefore, write the magnetic field inside the shell as

$$\vec{B}_{in} = \frac{\mu_0 m}{2\pi R^3}\left(\cos\theta\hat{r} - \sin\theta\hat{\theta}\right), \tag{A6}$$

and the solution outside is

$$\vec{B}_{out} = \frac{\mu_0 m}{2\pi r^3}\left(\cos\theta\hat{r} + \frac{\sin\theta}{2}\hat{\theta}\right), \tag{A7}$$

where we have defined the dipole moment, $m$, to be

$$m = \frac{\Omega Q R^2}{3}. \tag{A8}$$

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
