# Peer review of "Measuring Electromagnetic Fields in Rotating Frames of Reference"

_universe, doi:10.3390/universe6020031_

Round 1

Reviewer 1 Report

The manuscript fits well in the Special Issue in question, and I believe that it gives a usefull contribution to the issue of electromagnetic fields in rotating frames, particularly in what concerns practical and experimental questions.
I therefore recommend publication.

Besides the list of suggestions below, concerning mainly typos,
I would like to add the following suggestions to the authors:

I - If possible, please include more recent references.

II - Equation numbers, when refered to in the text, should probably include brackets, eg like Equation (66), instead of "equation 66".

III - The notation $\vec E_a$, etc in page 5 and elsewhere is not particularly nice, since it may be confused with coordinates. Furthermore,
in the appendix, $a$ and $b$ denote constants.

List of suggested corrections and typos (numbers refers to lines in the updated file)

73 "equal and opposite charge densities" . It is perhaps the total charge that is "equal", not the densities, since the shells have different radius.

90 "Ee" -> We?

128 typo "analysise"

130 "covariant Maxwell given in 3" -> "covariant Maxwell tensor given in (3)"?

146 There seems to be missing a reference to Schiff's paper [16].

163 There seems to be missing a reference to Crater's work, paper [4] I suppose.

190 the comma at (28) should probably be a period.

215 In (33) there is a missing equation.

219 in (35) the index should probably be $c$, instead of $b$

219 in (35) the period should probably be a comma

264 The meaning of the sentence
"It is important also to notice that $\hat \phi$ is not a relative velocity,
but describes an absolute rotation." is unclear to me

307 I suppose that the authors mean "point charge"

343 Possibly a missing "That" in "we find THAT the ... is"

355 typo "equations27and"

387 typo "49can"

396 missing $\vec$ and space in $B_s$

418-417 The sentence "... of an ’instantaneous’ of Lorentz transformations, ..." does not sound right.
There seems to be something missing

456 "spherical polars" maybe spherical polar coordinates

457 typo "$a$and $b$" instead of "$a$ and $b$"

464 "$sigma$is" instead of "$sigma$ is"

Reviewer 2 Report

This is an interesting paper showing explicitly how the definition of observable changes the results of experiment. Here the observable is the electromagnetic field produced by two rotating shells equiped with charges: Q and -Q. This field is measured in a rotating reference frame. In the first method, given many years ago by L.Schiff, the field is measured in the respective coordinate system. The second method proposed by authors as more realistic is realised in a suitable orthogonal tetrad with the spatial axes orthogonal to the world line of the observer. The authors show the difference between these two methods. The method of orthogonal tetrads is well known in both special and general relativity and it is exactly the mobile reper Cartan method well known in geometry. So the reviewed paper shows again the important role of the Cartan method in relativity.The only critical comment I have is the lack of some explanation of the matrices (39) and (40). Although authors refer a reader to Irvine's work but in my opinion it would be better to have any proof in the paper.

Concluding: I will  recommend the paper for publication after the improvement mentioned above is done. 
